# Yacon (*Smallanthus sonchifolius*) Flour Reduces Inflammation and Had No Effects on Oxidative Stress and Endotoxemia in Wistar Rats with Induced Colorectal Carcinogenesis

**DOI:** 10.3390/nu15143281

**Published:** 2023-07-24

**Authors:** Mariana Grancieri, Mirelle Lomar Viana, Daniela Furtado de Oliveira, Maria das Graças Vaz Tostes, Mariana Drummond Costa Ignacchiti, André Gustavo Vasconcelos Costa, Neuza Maria Brunoro Costa

**Affiliations:** Department of Pharmacy and Nutrition, Center for Exact, Natural and Health Sciences, Federal University of Espirito Santo, Alto Universitário, S/N Guararema, Alegre 29500-000, ES, Brazil; marianagrancieri@gmail.com (M.G.); mirellelomar@gmail.com (M.L.V.); danielafdo@hotmail.com (D.F.d.O.); mgvaztostes@gmail.com (M.d.G.V.T.); marianadci@gmail.com (M.D.C.I.); agvcosta@gmail.com (A.G.V.C.)

**Keywords:** prebiotics, yacon, endotoxemia, immune system, in silico, TNF-α

## Abstract

Colorectal cancer has a high worldwide incidence. The aim of this study was to determine the effect of yacon flour (YF) on oxidative stress, inflammation, and endotoxemia in rats with induced colorectal cancer (CRC). The Wistar male rats were divided and kept for 8 weeks in four groups: S (basal diet, *n* = 10), Y (YF flour + basal diet, *n* = 10), C (CRC-induced control + basal diet, *n* = 12), CY (CRC-induced animals + YF, *n* = 12). CRC was induced by intraperitoneal injections of 1,2-dimethylhydrazine (25 mg/kg body weight). Groups Y and CY received 7.5% of the prebiotic FOS from YF. The treatment with YF increased fecal secretory immunoglobulin A levels and decreased lipopolysaccharides, tumor necrosis factor alpha and interleukin-12. However, no effect was observed on the oxidative stress by the total antioxidant capacity of plasma, anion superoxide, and nitric oxide analysis of the animals (*p* < 0.05). The short-chain fatty acids acetate, propionate, and butyrate showed interactions with NF-κB, TLR4, iNOS, and NADPH oxidase by in silico analysis and had a correlation (by the Person analysis) with CRC markers. The yacon flour treatment reduced the inflammation in rats with induced CRC, and could be a promising food to reduce the damages caused by colorectal cancer.

## 1. Introduction

Colorectal cancer (CRC) has been associated with increased cancer-related mortality, since it the second cause of cancer-related death in the world [1]. Most cases of CRC are classified as sporadic, which involves mutations in the adenomatous polyposis gene, DNA hypomethylation, and multiple epigenetic changes. However, besides genetic factors, environmental and lifestyle risk factors are also involved [2]. Furthermore, genetic, pharmacological, and epidemiological data information showed an association between inflammation and CRC, contributing to its progression and development [3,4].

The intestine is constituted by some physical and immunological barriers that protect its integrity, which include epithelial cells that isolate the intestinal microbiota from the deeper gut tissue, as well as mucus, and immunoglobulin A (IgA). These factors prevent pathogens from translocating to blood circulation, which could start an inflammatory cascade, leading to chronic inflammation and the development of diseases, such as CRC [5]. However, in the presence of cancer cells, when the CRC has already taken hold, the inflamed colorectal epithelial cells do not constitute an effective barrier, allowing for the entrance of bacteria and their derivatives, such as lipopolysaccharides (LPS). LPS is the main constituent of the cell membrane of Gram-negative bacteria and acts as an endotoxin. LPS can bind with the Toll-like receptor 4 (TLR4), which leads to the activation of many downstream Mitogen-Activated Protein Kinases (MAPK), of which can induce cell proliferation, apoptosis, and adhesion, and induce the activation of the nuclear factor-kappa B (NF-κB) signal pathway [6]. NF-κB acts as a transcription factor responsible for the production of many pro-inflammatory cytokines, such as tumor necrosis factor alpha (TNF-α), interleukin-12 (IL-12), and IL-6 [2,6]; these increase the inflammatory process that increases and feeds the carcinogenesis process, forming a loop. 

Together with the high inflammation levels, cancer cells generate more reactive oxygen species (ROS), such as superoxide (O_2_^−^), hydroxyl radical (·OH), and hydrogen peroxide (H_2_O_2_), than normal cells. In general, a higher ROS level occurs in most cancer cells and their overproduction can intensify inflammation and intestinal barrier dysfunction, allowing for the translocation of bacteria and toxins. Furthermore, ROS production can drive cell injury and apoptosis by alterations in lipids, proteins, and DNA, and is a major cause of tumorigenesis [1,7].

The oxidative stress, as well as inflammation, may be regulated by some nutritional components [8]. Furthermore, there is a large amount of evidence stating that dairy products, whole grains, and dietary fiber consumption have a protective effect against CRC development [9]. Yacon (*Smallanthus sonchifolius*) is an Andean root that contains 40% to 70% of fructooligosaccharides (FOS) (dry matter), which is considered a prebiotic and is associated with CRC prevention [10,11].

It has been demonstrated that treatment with yacon flour increases the short-chain fatty acid (SCFA) levels, improves the intestine architecture, reduces aberrant crypts focus in rats with CRC [11,12,13] and also reduces the intestinal permeability and luminal pH [11,14]. Furthermore, rats with CRC treated with yacon flour for 2 weeks were able to increase their fecal IgA levels [15]. Therefore, the use of rats in this experimental model is well established; the immunological benefits of yacon intake have been demonstrated in other studies, in which serum anti-inflammatory cytokines, such as interleukin (IL)-10 and IL-4, were increased [16,17], and pro-inflammatory cytokines, such as interferon γ (IFNγ), were decreased. In addition, yacon consumption increased antioxidant enzymes [18].

Therefore, considering the positive feedback between CRC, oxidative stress and inflammation and its relationship with the malefic effects of CRC, it is necessary to establish new food alternatives to combat cancer and its complications. The aim of this study was to determine the effect of yacon flour (YF) on oxidative stress, inflammation, and endotoxemia in rats with induced colorectal cancer (CRC). Our hypothesis is that yacon flour, as a rich source of prebiotic fructooligosaccharide, can reduce inflammation, endotoxemia, and oxidative stress caused by colorectal cancer.

## 2. Methods and Materials

### 2.1. Animals and Experimental Diet

The male Wistar rats (*n* = 46) were from the Central Animal Breeding of the Universidade Federal do Espírito Santo, with 207 ± 5 g of initial body weight. The animals were kept in a room with control conditions (22 ± 2 °C, 12 h light–dark cycle) and water ad libitum. All the experimental procedures were performed in compliance with the ethical principles for animal experimentation, in accordance with Directive 86/609/EEC of 24 November 1986. The “Ethics Committee of Animals Use” from Universidade Federal do Espírito Santo (No. 004/2014) approved the study.

### 2.2. Yacon Flour Preparation and Experimental Diet

The yacon roots from Santa Maria de Jetibá, ES, Brazil, were from the same lot and prepared according to Vaz Tostes et al. [16]. All chemical analyses were performed in triplicate using the AOAC methods [19]. Moisture was determined using an oven (Nova Ética^®^, model 400/6 ND, São Paulo, Brazil) at 105 °C, ash by a muffle furnace (Quimis, Q320 M model, Diadema, SP, Brazil) at 550 °C, protein content by the Kjeldhal method, and lipid content through the Soxhlet method. Total dietary fiber was determined by the gravimetric–enzymatic method using a commercial kit (total dietary fiber assay kit, Sigma^®^, Sigma-Aldrich, Barueri, SP, Brazil). The yacon flour calories were determined using a numerical calculation based on the macromolecule composition.

The content of FOS, inulin, and simple carbohydrates (glucose, fructose, and sucrose) in the yacon flour were identified and quantified by High Performance Liquid Chromatography (HPLC); column HPX-87p (BIO-RAD Laboratories, Santo Amaro, SP, Brazil); mobile phase: purified water [11]. The yacon flour composition is shown in Table 1.

The experimental diet was based on the AIN-93M diet [20]. The diet of groups Y and CY was supplemented with 7.5% of FOS from yacon flour (14.37 g of YF/100 g of diet). Considering the yacon composition, all diets were adjusted to present analogous amounts of proteins, fibers, simple carbohydrates, and calories, so that the only additional nutrient was FOS from yacon flour (Table 2). Diets were stored (4 °C) for 15 days (maximum) to avoid FOS degradation.

### 2.3. Experimental Design

Out of 46 animals, 20 were kept healthy and 26 animals were induced to CRC through a subcutaneous injection (25 mg/kg body) of DMH (1,2-dimethylhydrazine, Sigma^®^) once a week for five weeks. DMH (pH 6.5) was prepared immediately before use by dissolution in NaCl (0.9%) and EDTA (15%) [21]. The subsequent 8 weeks consisted of an interval for CRC development.

At the end of the CRC induction period, in the 13th week of the experiment, two animals with CRC induction were euthanized for confirmation of aberrant crypt foci (ACF) development [22]. The other 24 animals with induced CRC were randomly divided by weight into groups C and CY, and the 20 animals without induced CRC (healthy) were randomly divided by weight into groups S and Y. Then, each group received their respective diets:-Group S: animals without colorectal cancer induction and without yacon flour; *n* = 10-Group Y: animals without colorectal cancer induction and with yacon flour; *n* = 10-Group C: animals with colorectal cancer induction and without yacon flour; *n* = 12-Group CY: animals with colorectal cancer induction and with yacon flour; *n* = 12

During the first 13 weeks (CRC-induction), all animals were fed with a basal and commercial diet (Brand: In Vivo™). In the 8 following weeks, the animals of groups S and C received the AIN-93M diet, and Y and CY groups received the AIN-93M diet with an addition of yacon flour to provide 7.5% of FOS (Grancieri et al. 2017). Each animal remained in an individual cage; they were properly identified to avoid errors between groups and animals, and all the researchers knew the identification of the animals.

At the end of experiment (22nd week), the animals were anesthetized (0.2 mL/100 g body weight) with a solution containing 37.5% ketamine, 25% xylazine, and 37.5% of the saline solution by intraperitoneal administration, and then the blood was collected by a cardiac puncture. The blood was placed in tubes with sodium heparin to obtain plasma, and in polypropylene tubes (Falcon, Fisher Scientific^®^, São Paulo, SP, Brazil) to evaluate the whole blood for immune analysis. The plasma was obtained by centrifugation of the blood (200× *g*, 10 min, 4 °C), separated from the erythrocyte part, and frozen at −80 °C. In addition, the luminal contents of the large intestines of the animals were collected and frozen at −80 °C (Figure 1).

### 2.4. Oxidative Stress Markers

The preparation of opsonized zymozan-A and isolation of neutrophils was carried out according to Henson [23] using zymozan-A from Saccharomyces cerevisiae, which was resuspended in rat serum (from extra animals), and the concentration was adjusted to 10 mg/mL with Hanks Hepes.

The neutrophils were isolated from blood with 2.5% gelatin in PBS. The final cell concentration was adjusted to 5 × 10^5^ cells/mL in Hanks Hepes (15 mM, pH 7.2, 0.1% gelatin). The extracellular release of superoxide anion (O_2_^−^) was measured by the ferrocytochrome C reduction method, considering only the inhibitory reduction by superoxide dismutase (SOD), as previously described [24]. The final suspension was read at 550 nm in length, and the resulting optical density was converted to reduced ferrocytochrome C using ΔE550 nm = 2.1 × 10^4^ M^−1^·cm^−1^.

Regarding the nitric oxide production determination, 100 μL of the culture supernatant was mixed with the same volume of the Griess reagent (ref#G4410, Sigma^®^) and incubated at room temperature for 10–15 min [25]. Absorbance was determined at 540 nm. The NO_2_^−^ concentration was determined using a standard curve (0.2–100 µM) of sodium nitrite (NaNO_2_).

### 2.5. Secretory IgA, Endotoxemia and Total Antioxidant Capacity of Plasma

The concentration of sIgA was determined in the luminal contents of the animals’ colons by the ELISA method, following the commercial kit instructions (Sigma-Aldrich^®^, St. Louis, MO, USA). The concentration of IgA in the samples was determined from a standard curve (0–200 μg/dL), and the results were given in μg/dL.

Endotoxemia was measured by the lipopolysaccharide (LPS) levels in the plasma. The analyses were carried out using the Limulus amoebocyte lysate (LAL) test by a commercial kit (Hycult Biotech^®^, HIT302, Hycult Biotech, Uden, The Netherlands) and standard LAL curve (0.04–10 EU/mL), following the manufacturer’s recommendation. The results were expressed in EU/mL (endotoxin unit per mL plasma).

The TAC (total antioxidant capacity of plasma) was performed using a colorimetric kit (Cayman Chemical Companyl^®^, Ann Arbor, MI, USA), with Trolox (6-hydroxy-2,5,7,8-tetramethylchroman-2-carboxylic acid) as a standard. The results were expressed as millimoles (mM) of the Trolox equivalent (TE).

### 2.6. Cytokines Release

The blood was centrifuged (3000 g, 10 min, 4 °C) and the obtained plasma was analyzed for cytokine determination. Commercial kits were used to quantify IL-10 and IL-12 cytokines (Milliplex^®^ Map) and TNF-α (EMD Millipore^®^, São Paulo, SP, Brazil) by the ELISA methods. The results were expressed in pg/mL from a standard curve of IL-12 (0–50,000 pg/mL), IL-10 (0–30,000 pg/mL), and TNF-α (0–30,000 pg/mL).

### 2.7. Aberrant Crypt Foci (ACF) Analysis

The colon of animals was fixed in formalin and stained with 1% methylene blue. The ACF on the mucosal surface of the large intestine were counted by two independent and trained researchers in a blind manner, using an optical microscope (4X objective).

### 2.8. Intestinal Permeability and Fecal Short-Chain Fatty Acids (SCFAs) Analysis

In the last week of experiment, the animals were fasted for 12 h and then received, by gavage, 2 mL of the solution (200 mg of lactulose and 100 mg of mannitol); samples of their urine were collected for 24 h. At the end, the collected urine volume was measured, recorded and stored at −80 °C [26]. Then, for the analysis, the urine was diluted 1:2 with distillated water, filtered (0.45 µm) and analyzed by the HPLC method.

Furthermore, the colonic feces of animals were quantified for their levels of SCFA, acetate, propionate, and butyrate. The extraction of the SCFA was performed by mixing 100 mg of luminal contents with 2 mL of perchloric acid (10%) and centrifuged (9000× *g*, 10 min, 25 °C). The supernatant was filtered (0.45 µm) and analyzed by HPLC [27].

The HPLC conditions for the analyses of lactulose, mannitol and SCFA were a Shimadzu HPLC system (Kyoto, Japan) with a degasses (Model DGU-14A), pump (Model LC-10AT), auto-sampler (Model SIL-20A), column oven (Model CTO-10AS), UV–vis detector (Model SPD-10AV), and a refractive index detector (Model RID-10A). The column used was Aminex HPX-87H (300 cm × 8.7 mm; BIO-RAD) in 55 °C with a pressure of 1920 psi, using H2SO4 0.005 mM as a mobile phase under isocratic conditions [28]. Lactulose, mannitol and SCFA levels (mg/g of feces) were determined by standard curves using commercial standards (Sigma-Aldrich).

### 2.9. Intraluminal pH of the Colon

The cecal luminal contents were removed, weighed, and 4 mg was diluted in 400 mL of distillated water. After the complete homogenization by vortex, the pH was read using a pH meter (Kasvi^®^, São José dos Pinhais, PR, Brazil) [11].

### 2.10. In Silico Analysis

The structural interaction between short-chain fatty acids (SCFAs) acetate, propionate, and butyrate, and p65-NF-κB, TLR4, inducible Nitric Oxide Synthase (iNOS), and Nicotinamide Adenine Dinucleotide Phosphate Oxidase (NADPH oxidase) was evaluated by molecular docking. The SCFAs were designed using Instant MarvinSketch (ChemAxon Ltd., Boston, MA, USA). The crystal structure file of p65-NF-kB, TLR4, iNOS, and NADPH oxidase was obtained from the Protein Data Bank (http://www.rcsb.org/, accessed on 16 January 2023) (PDB: 1OY3, 3FXI, and 3E6T, 1HH4, respectively).

Non-polar hydrogen atoms were merged, and rotatable bonds were defined on the AutoDockTools^®^ program. Flexible torsions, charges, and the grid size were assigned by AutoDock Tools [29], and the docking calculations were performed using AutoDock Vina [30]. The active site of target enzymes was based on an exhaustive search in the literature of studies which demonstrated the binding sites of the enzymes; then, based on these results, successive tests were carried out to locate the best active site of the enzymes. The binding pose with the lowest binding energy (highest binding affinity) was selected as a representative image to visualize in the Discovery Studio 2016 Client (Dassault Systèmes Biovia Corp^®^, Hudson, OH, USA) [31].

### 2.11. Statistical Analysis

The power analysis to determine the sample size was performed, as indicated by Lwanga [32], for analytical studies using α = 5% and z α/2 = 1.96, as used in health studies, accomplished using our previous study as a basis [11]. The statistical analysis procedures were conducted with software GraphPad Prism, version 9.0. The data normality were tested by the Kolmogorov–Smirnov test. The normal data were analyzed by ANOVA and the post hoc Newman–Keuls method. The Correlation Matrix analysis was carried out using the Person test. Data were expressed as means ± standard deviation (SD), using *p* < 0.05 as the level of significance.

## 3. Results

### 3.1. Yacon Composition

It was observed that yacon flour (in the dry basis) is a rich source of prebiotics, mainly FOS and inulin. Furthermore, we observed the presence of other fibers in its composition. The simple carbohydrates were the main macronutrient present, and we observed the presence of fructose, sucrose and glucose. Additionally, proteins and ash were identified; however, the amount of lipids was very small (Table 1).

### 3.2. Superoxide Anion and Nitric Oxide Release

It was observed that none of the groups, treated or not with yacon flour and with or without induced colon cancer, differed in the release of superoxide anion, either in the unstimulated (Figure 2A) or stimulated (Figure 2B) neutrophils with opsonized zymozan (*p* > 0.05).

Likewise, all groups showed similar nitric oxide secretions, regardless of whether they were stimulated by zimozan or not (Figure 2C,D) (*p* > 0.05).

### 3.3. Secretory IgA Production, Endotoxemia and Total Antioxidant Capacity

The Y group had the highest levels (*p* < 0.05) of sIgA, and the values were similar to the CY group, which also received yacon flour and had colorectal cancer induction. The C group had the lowest IgA secretion values (*p* < 0.05). The S group did not present statistically different values when compared to the Y, C, and CY groups (Figure 3A).

Groups with induced colorectal cancer, the C and CY groups, showed the highest endotoxemia values (*p* < 0.05). However, the CY group had similar values to the S and Y groups, which had no induction of colorectal cancer. Furthermore, the S and Y groups showed similar and lower values of endotoxemia (Figure 3B).

In addition, all groups, regardless of treatment, had similar values for total plasma antioxidant capacity (Figure 3C).

### 3.4. Cytokines Release

The C group showed the highest plasma levels of pro-inflammatory cytokines TNFα (Figure 4A) and IL-12 (Figure 4B) (*p* < 0.05). The CY group had the lowest levels of those cytokines (*p* < 0.05) followed by the C group. The release of TNFα and IL-12 were similar between S, Y, and CY groups (*p* > 0.05) (Figure 4A,B).

However, the values of anti-inflammatory cytokine IL-10 were similar between S, Y and C groups. The CY group had the lowest plasma levels of this cytokine (*p* < 0.05) (Figure 4C).

### 3.5. In Silico Analyses

Butyrate showed the highest interaction by lowest estimated free energy (EFE) with p65-NF-κB, iNOS, and NADPH oxidase (EFE of −3.4, −4.3, and −4.1, respectively), whereas acetate showed the lowest interaction with these markers. Propionate showed the highest interaction with TLR4 (EFE of −3.6) and butyrate showed the lowest interaction with TLR4 (EFE of −3.1) (Table 3 and Figure 5).

### 3.6. Correlation Analysis

It was observed that the urinary excretion of mannitol and lactulose had a positive correlation with each other (*p* < 0.05). In addition, both showed a positive correlation with fecal pH and IL-10. The mannitol had also a positive correlation with TNF and IL-12 (*p* < 0.05). On the other hand, these sugars showed an inverse correlation with the levels of propionate and butyrate in the feces (*p* < 0.05). The fecal pH showed an inverse correlation to the levels of propionate and IgA: the lower the pH, the higher the value observed of IgA and propionate in the feces. Conversely, the ACF values showed a significant inverse correlation only with acetate, and TAC had a significant positive correlation with butyrate in the feces (*p* < 0.05). Acetate also had an inverse correlation with serum LPS and a positive correlation with the anti-inflammatory cytokine IL-10 (*p* < 0.05). Propionate showed a positive correlation with sIgA and butyrate in the feces. The levels of the inflammatory cytokine IL-12 showed, in turn, a positive correlation (*p* < 0.05) with LPS and urinary mannitol (Figure 6).

## 4. Discussion

This study demonstrated that the yacon flour, as a source of FOS, was able to reduce inflammatory biomarkers in CRC-induced animals. Chronic inflammation can be modulated by some nutritional compounds, such as prebiotics. Prebiotics are non-digestible food components that modulate gut microbiota by stimulating the growth and/or activating the metabolism of probiotic bacteria, such as Lactobacillus and Bifidobacterium spp. [33].

Yacon (*Smallanthus sonchifolius*) flour is a rich source of prebiotics by providing 52.2% of FOS, 6.61% of inulin, and 10.68% of other fibers (Table 1). The FOS, as well as being associated with probiotics, can modulate the immune resistance [34] in addition to providing protective effects against early biomarkers and the development of tumors in the colon of rats [11,12,14]. Furthermore, FOS can be fermented by gut microbiota, generating SCFAs, mainly acetate, butyrate, and propionate, which are associated with health benefits, including anti-cancer effects [35]; this was shown in our results, whereby pH was inversely the production of acetate and inversely related with the ACF formation. The fermentation was confirmed since the pH was inversely correlated with propionate, i.e., the fermentation reduced the pH by the production of SCFAs, confirming our previous works [11,14]. Furthermore, the reduction in ACF, related with acetate, was also confirmed in our previous work [11]. ACFs are the first lesions detected microscopically in colorectal carcinogenesis; they are true preneoplastic lesions, commonly used as an indicator of the early stages of the disease, and its reduction indicates a potential to suppress CRC [36].

The healthy gut microbiota has a key role in the induction of defense cell production, such as T cells and secretory immunoglobulin A (sIgA) [37]. In this study, the yacon flour improved the sIgA levels even in animals with CRC (CY group). This result may be due to microbiota modulation by prebiotic FOS, which stimulates the B lymphocyte to produce sIgA [37,38,39]; this was confirmed by the positive association between sIgA and propionate, which confirmed that the fermentation induced by yacon has the ability to induce sIgA. This immunoglobulin is predominant in secretions such as saliva, tears, colostrum, and feces; this immunoglobulin does not bind complement, and may act against microorganisms without triggering a progressive inflammatory process that damages the epithelium. In addition, sIgA acts by neutralizing toxins and pathogenic microorganisms, preventing them from binding or passing through the intestinal mucosa [40]. Other studies also observed an increase in sIgA levels after yacon ingestion in experimental studies [15,37,41] and human trials [16].

Furthermore, in the current study, the endotoxemia levels on CRC-induced animals treated with yacon flour (CY) were similar to those seen in healthy animals, which can contribute to the reduced inflammation process, as confirmed by the lower IL-12 and TNF-α levels, the major cytokines of the inflammation process [42], thus confirming our original hypothesis. These results are further supported by the verification that a higher excretion of mannitol is positively related to the values of inflammatory cytokines IL-12 and TNF by correlation analysis. Then, yacon consumption possibly reduced serum inflammation by reducing intestinal permeability, which reduces bacterial translocation (by reducing LPS release) and the induction of inflammatory cytokines.

Under normal conditions, the LPS in the intestinal lumen does not cause negative effects. However, some factors, such as dysbiosis, fat consumption, and obesity, may favor the transfer of LPS to the circulatory system, a process called endotoxemia [1,2]. Then, the intracellular signaling pathways and transcription factors such as NF-κB are activated in the plasma cells, such as neutrophils, macrophages, and lymphocytes, which release pro-inflammatory cytokines [43].

The low TNF-α levels observed can be associated with the suggested capacity of SCFAs to bind into NF-κB and TLR4, as proposed in the in silico analysis, which can reduce the activation of these biomarkers and, consequently, their associated pathways. Other studies observed a reduction in endotoxemia after xylo-oligosaccharide and inulin consumption by healthy patients [17] and by mice fed with a high-fat diet [44]. Nevertheless, the levels of anti-inflammatory IL-10 were not increased as expected, although acetate levels positively correlated with IL-10 levels. This result was probably due to the systemic evaluation instead of local IL-10 levels, as in the intestine, this cytokine is released by macrophages, dendritic cells, and epithelial cells [45,46]. A study with preschool children also did not find an increase in IL-10 levels associated with yacon flour intake, despite the increased levels of IL-4 [16].

We observed favorable interactions of the ligand-proteins in every in silico analysis, such as hydrogen bonding, the carbon–hydrogen bond, and Pi-sigma bond; they can contribute to lowering the system’s free energy, especially the hydrogen bonds that play an essential role in protein–ligand binding [47,48]. Despite the observed supposed interaction between SCFAs and NADPH oxidase and iNOS by in silico analysis, we do not observe the results in oxidative stress between groups, independently of stimulus, in nitric oxide and superoxide levels. These results indicate that both cancer and yacon flour had no impact on these oxidative and inflammatory markers, the opposite of what we initially hypothesized for the study. Furthermore, a possible explanation of why cancer did not increase the oxidative stress is because the cancer was already present. This disease, to avoid cytotoxic signaling and to facilitate tumorigenic signaling, has a local mechanism in place that monitors ROS, such as signaling molecules that convey information about increases in oxidative stress to the nucleus in order to upregulate antioxidant genes [49].

High levels of ROS are associated with DNA damage, significant toxicity, cell apoptosis, endothelium dysfunction, among others, which have been implicated in many diseases, including cancer [50]. Healthy mice fed with FOS from yacon did not demonstrate any alteration on NO levels [51]. Another similar study which analyzed the effects of yacon flour on the prevention of CRC also found no changes in the antioxidant capacity of animals [14].

Those results were confirmed by the similarity of the TAC values between the groups, confirming that no oxidative stress was induced in the animals, regardless of the tested treatment. The organism had mechanisms to fight against ROS production, which include endogenous enzymes, like glutathione-peroxidase, superoxide dismutase, paraoxonase-1, catalase. In addition, yacon flour is a rich source of antioxidant compounds, such as phenolics, chlorogenic, caffeic, coumaric and protocatechuic acids [52], which can improve the oxidative stress; however, in this study, this effect was not observed. The TAC analyses provide more relevant biological information than individual components, once the cumulative effect of all antioxidants present in body fluids and plasma is considered [53].

The proposed effect of yacon flour in our study is that its consumption, as a rich source of FOS, increased fecal sIgA, improving the barrier that prevents pathogen bacteria from passing through the intestinal membrane, contributing to the reduction in endotoxemia. Such results directly contributed to the reduction in inflammatory cytokine secretion. These results may be related to the in silico interaction of the SCFAs produced by the fermentation of FOS with biomarkers of inflammation. However, despite the in silico effect of SCFAs on oxidative stress markers, none of the modifications to the antioxidant were observed in animals, which may have occurred due to the non-induction of oxidative stress by colorectal cancer. However, we emphasize that these connections between SCFAs and enzymes, demonstrated in the in silico analysis, are plausible to occur, but require caution in their interpretation, since in a physiological context, other factors can interfere in these connections and in the consequent actions of the enzymes [54]. The main limitation of this study was the lack of studies on the same topic, as it made it difficult to establish the exact dose of yacon flour, the treatment time and the main markers for assessing oxidative stress (Figure 7).

## 5. Conclusions

We confirm our hypothesis that yacon flour improves the inflammation caused by colorectal cancer; however, we have not observed the effects on oxidative stress and endotoxemia. These results are innovative and highlight the effectiveness of yacon flour against inflammation in CRC-induced rats. Furthermore, this study could drive further studies to investigate the molecular mechanisms related to these systemic effects observed in CRC-induced rats, as well as clinical investigations.

## Figures and Tables

**Figure 1 nutrients-15-03281-f001:**
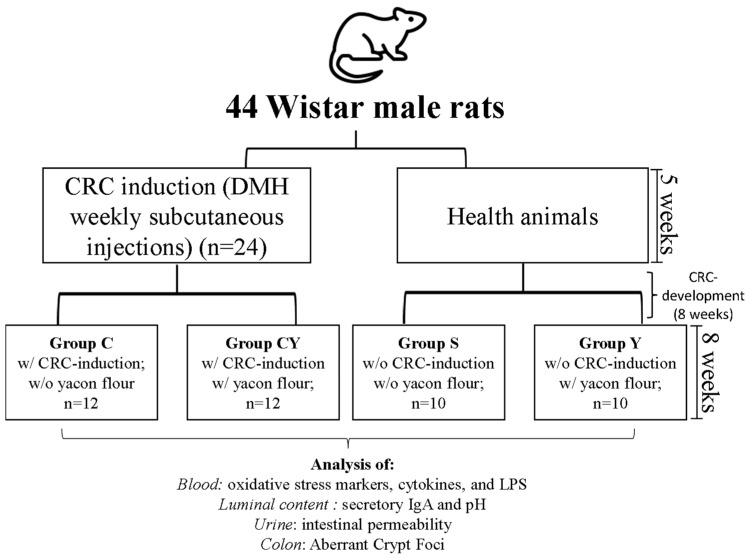
Experimental designer of experiment. DMH: 1,2-dimethilhydrazine; CRC: colorectal cancer; w/: with; w/o: without.

**Figure 2 nutrients-15-03281-f002:**
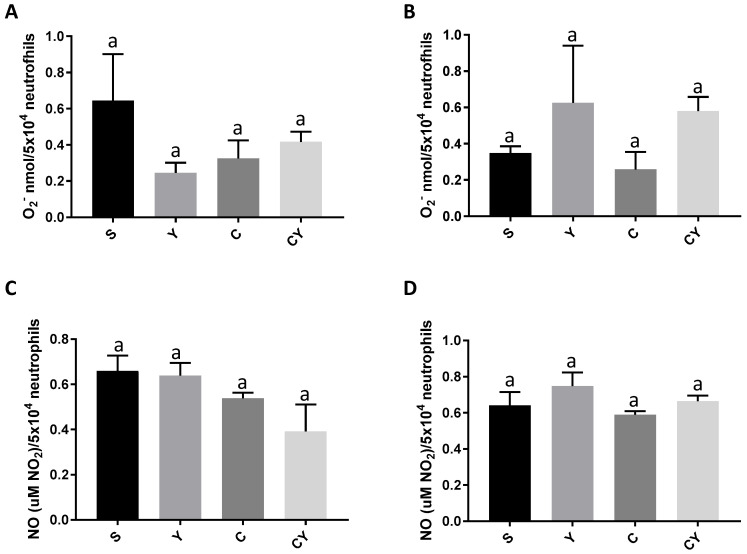
Superoxide anion and nitric oxide release by neutrophils from rats fed or not with yacon flour and induced or not to colorectal cancer. (**A**) Superoxide anion release by neutrophils not stimulated by opsonized zymozan; (**B**) superoxide anion release by neutrophils stimulated by opsonized zymozan; (**C**) nitric oxide release by neutrophils not stimulated by opsonized zymozan; (**D**) NO release by neutrophils stimulated by opsonized zymozan. Values are shown as means ± SD. Dates analyzed by one-way ANOVA and post hoc Newman–Keuls (*p* < 0.05). The lowercase letter “a” indicates that there is no difference between groups. Dates analyzed by absorbance. S = group without the induction of colon cancer and without yacon flour (*n* = 10); C = group with the induction of colon cancer and without yacon flour (*n* = 10); Y = group without the induction of colon cancer and with yacon flour (*n* = 12); CY = group with the induction of colon cancer and with yacon flour (*n* = 12). YF = yacon flour; CRC = colorectal cancer. O_2_^−^: superoxide anion; NO: nitric oxide.

**Figure 3 nutrients-15-03281-f003:**
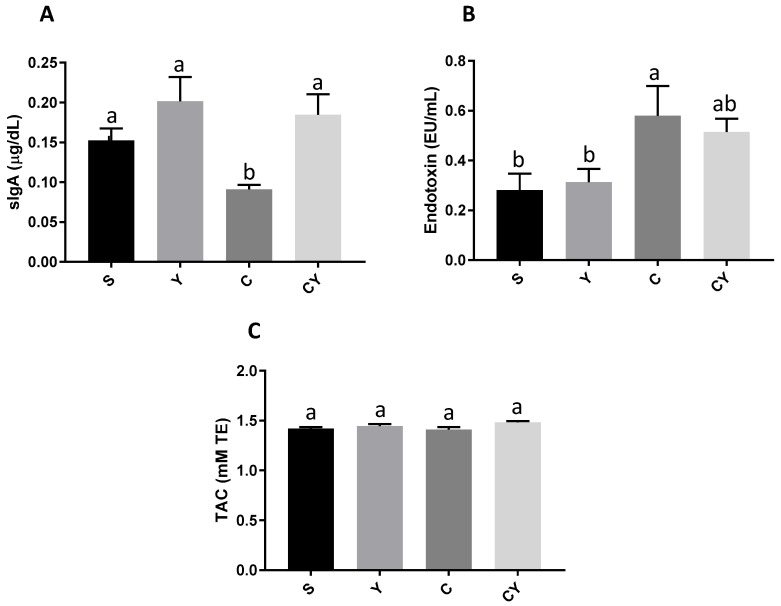
Immunologic and oxidative markers produced by rats fed or not with yacon flour and induced or not to colorectal cancer. (**A**) IgA release; (**B**) endotoxin levels; (**C**) TAC. Values shown as means ± SD. Dates are analyzed by one-way ANOVA and post hoc Newman–Keuls (*p* < 0.05). The different letters are the differences between groups. Dates analyzed by absorbance. S = group without induction of colon cancer and without yacon flour (*n* = 10); C = group with induction of colon cancer and without yacon flour (*n* = 10); Y = group without induction of colon cancer and with yacon flour (*n* = 12); CY = group with induction of colon cancer and with yacon flour (*n* = 12). YF = yacon flour; CRC = colorectal cancer. sIgA= secretory immunoglobulin A. TAC= total antioxidant capacity. mMTE= millimolar of Trolox equivalent. EU/mL: endotoxin unit/mL plasma.

**Figure 4 nutrients-15-03281-f004:**
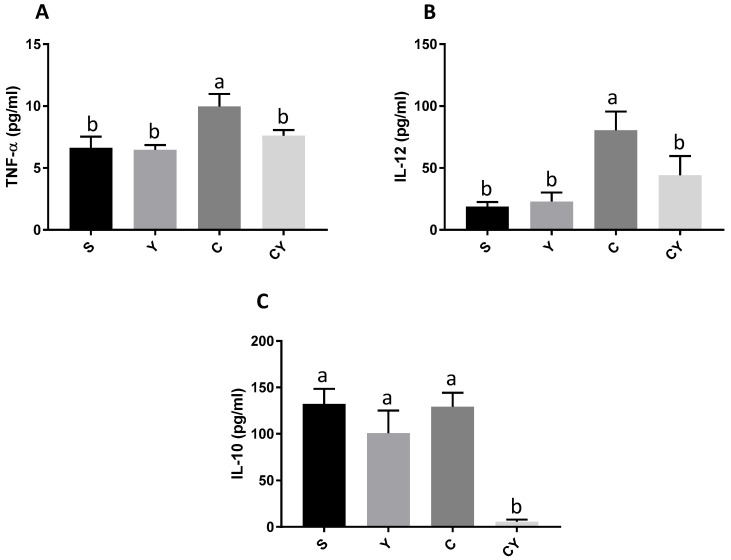
Plasmatic cytokine release by rats fed or not with yacon flour and induced or not to colorectal cancer. (**A**) TNF-α release; (**B**) IL-12 release; (**C**) IL-10 release. Values showed as means ± SD. Dates analyzed by one-way ANOVA and post hoc Newman–Keuls (*p* < 0.05). The different letters are the differences between groups. Dates analyzed by absorbance. S = group without induction of colon cancer and without yacon flour (*n* = 10); C = group with induction of colon cancer and without yacon flour (*n* = 10); Y = group without induction of colon cancer and with yacon flour (*n* = 12); CY = group with induction of colon cancer and with yacon flour (*n* = 12). TNF-α = tumoral necrosis factor alpha; IL-12 = interleukin-12; YF = yacon flour; CRC = colorectal cancer.

**Figure 5 nutrients-15-03281-f005:**
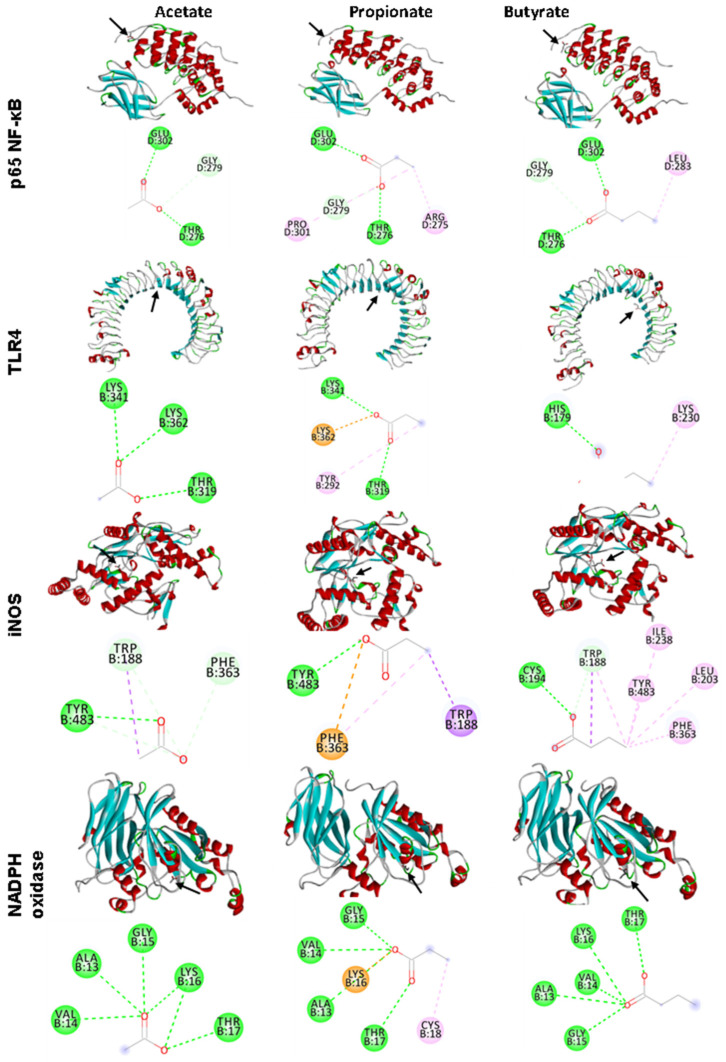
The in silico interaction of the main short-chain fatty acids generate by FOS fermentation after yacon consumption: acetate, propionate, and butyrate. Dates analyzed by AutoDock Vina^®^ and visualized by Discovery Studio 2016 Client^®^. Color code indicates the residue interaction: heliotrope: Pi-sigma bond; lime green: conventional hydrogen bond; light green: carbon hydrogen bond; neon pink: Pi-Pi T-shaped bond; orange: Pi-cation bond. p65 NF-κB: nuclear factor kappa B. TLR4: Toll-like receptor 4. iNOS: inducible nitric oxide synthase. NADPH oxidase: nicotinamide adenine dinucleotide phosphate oxidase.

**Figure 6 nutrients-15-03281-f006:**
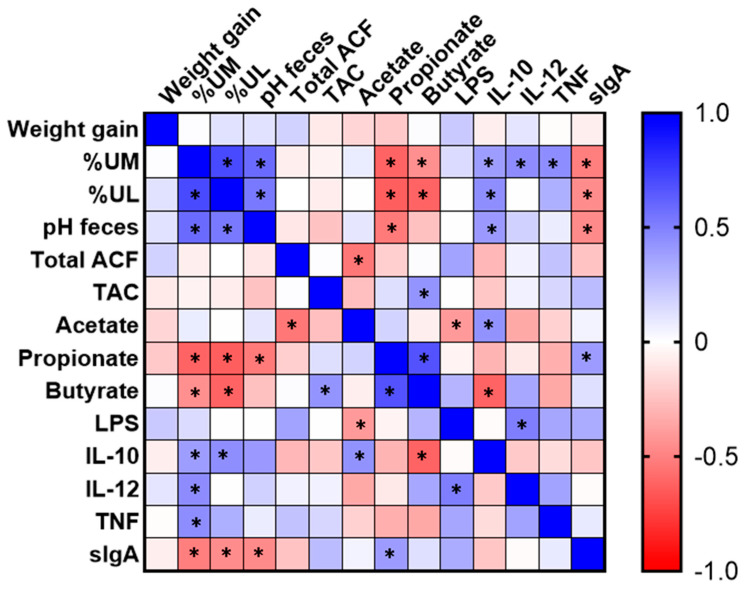
Heat map of the Pearson rank correlations between the biological outcomes of animals with induced colorectal cancer and fed with yacon flour for 8 weeks. * *p* < 0.05. %UM: percentage of urinary excretion of mannitol; %UL: percentage of urinary excretion of lactulose; total ACF: total aberrant crypt foci; TAC: total antioxidant capacity; LPS: lipopolysaccharide; IL: interleukin; TNF: tumor necrosis factor; sIgA: secretory immunoglobulin A.

**Figure 7 nutrients-15-03281-f007:**
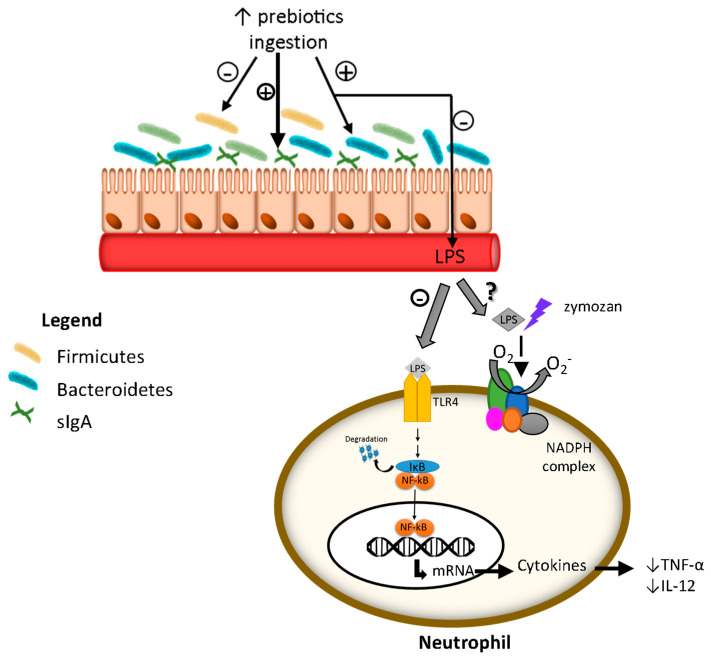
The proposal effect caused by the consumption of yacon flour as a FOS (prebiotic) source. The prebiotic ingestion stimulates the *Bacteroidetes* bacteria development instead of *Firmicutes* bacteria, which reduce the production and release of LPS to the blood circulation. Then, the binding of LPS on TLR4, and consequently the downstream pathway activation, is reduced on immune cells, like neutrophil, which reduces the cytokines’ release and reduces the inflammation. LPS: lipopolysaccharide; O_2_^−^: superoxide anion; NADPH oxidase: nicotinamide adenine dinucleotide phosphate oxidase; sIgA: secretory immunoglobulin A. TLR4: Toll-like receptor 4; p65 NF-κB: nuclear factor kappa B; IκB: inhibitor of κB; TNF-α = tumoral necrosis factor alpha; IL-12 = interleukin-12.

**Table 1 nutrients-15-03281-t001:** Composition of yacon flour (in the dry basis).

Components	Amount (g)
Non-digestible carbohydrates	
Fructooligosaccharide	52.20 ± 0.01
Inulin	6.61 ± 0.00
Other Fibers	10.68 ± 0.08
Macronutrients	
Simple Carbohydrates	
Fructose	8.16 ± 0.01
Glucose	3.76 ± 0.00
Sucrose	7.25 ± 0.01
Proteins	4.52 ± 0.25
Moisture	3.72 ± 0.51
Ash	2.94 ± 0.03
Lipids	0.33 ± 0.01

Data are presented as means ± standard deviation.

**Table 2 nutrients-15-03281-t002:** Composition of the AIN-93M diet with and without supplementation with yacon flour.

Ingredients (per kg of Diet)	S and C Groups	Y and CY Groups
	AIN-93M	AIN-93M + YF
Casein (g)	140.0	130.14
Dextrinized starch (g)	150.5	150.5
Sucrose (g)	100.0	70.24
Soybean oil (mL)	40.0	40.0
Microcrystalline cellulose (g)	50.0	0
Minerals Mix (g)	30.5	30.5
Vitamin Mix (g)	10.0	10.0
L-cystine (g)	1.80	1.80
Choline bitartrate (g)	2.50	2.50
Corn starch (g)	474.7	423.95
Yacon flour (g)	0	140.37
Nutrition composition		
Caloric Density (kcal/g)	3.72	3.54

S = group without colon cancer induction and without yacon flour; C = group with colon cancer induction and without yacon flour; Y = group without colon cancer induction and with yacon flour; CY = group with colon cancer induction and yacon flour. YF = yacon flour.

**Table 3 nutrients-15-03281-t003:** Estimated free energy binding and chemical interactions among the main short-chain fatty acids from FOS fermentation and the catalytic site of the p65NF-κB, TLR4, iNOS, and NADPH oxidase.

	Acetate		Propionate		Butyrate	
	EFE	IAAR	EFE	IAAR	EFE	IAAR
p65 NF-KB	−2.9	GLU D: 302; GLY D: 279; THR D: 276	−3.2	GLU D: 302; GLY D: 279; PRO D: 301; THR D:276; ARD D: 275	−3.4	TRH D: 276; GLY D: 276; GLU D: 302; LEU D: 283
TLR4	−3.4	LYS B: 341; LYS B: 362; THR B: 319	−3.6	LYS B: 341; LYS B: 362; THR B: 319; TYR B: 292	−3.1	HIS B: 179; LYS B: 230
iNOS	−3.2	TYR B: 483; TRP B: 188; PHE B: 363	−3.9	TYR B: 483; TRP B: 188; PHE B: 363	−4.3	CYS B: 194; TRP B: 188; ILE B: 238; TYR B: 483; LEU B: 203; PHE B: 363
NADPH oxidase	−3.6	VAL B: 14; ALA B: 13; GLY B: 15; LYS B: 16; THR B: 17	−4	GLY B: 15; VAL B: 14; LYS B: 16; ALA B: 13; THR B: 17; CYS B: 18	−4.1	GLY B: 15; ALA B: 13; VAL B: 14; LYS B: 16; THR B: 17

EFE: Estimated free energy (kcal·mol^−1^). IAAR: Interacting amino acid residues. Docking calculations were carried out using AutoDock Vina. Negative values mean spontaneous reaction. The most potent interaction between short-chain fatty acids and receptor is in bold.

## Data Availability

Not applicable.

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
