# Peer review of "Yacon (Smallanthus sonchifolius) Flour Reduces Inflammation and Had No Effects on Oxidative Stress and Endotoxemia in Wistar Rats with Induced Colorectal Carcinogenesis"

_nutrients, 2023, doi:10.3390/nu15143281_

Round 1

Reviewer 1 Report

This manuscript demonstrates that YF, a source of fructooligosaccharides, can reduce inflammatory biomarkers in CRC-induced animals. And the test results confirmed that YF could increase the secretion of sIgA and speculated the interaction between SCFA and NADPH oxidase and iNOS through in silico analysis. Although the animal experiments in the manuscript are adequately designed, and the text is clear and easy to read, some data are not measured or presented, so the credibility of the experimental results of the manuscript is slightly weak. For example, how much SCFA is produced by YF treatment? It is too exaggerated to use SCFA as an enzyme docking analysis to speculate on reducing cancer biomarkers. Some revisions will be more suitable for publication.

Main recommendations:

1. In Figure 3, the endotoxin of the CY group is compared with that of the C, Y, and S groups, which can be regarded as ambiguous. Extrapolating from this data that consumption of yacon may reduce serum inflammation by reducing intestinal permeability, bacterial translocation (by reducing lipopolysaccharide release), and inducing inflammatory cytokines, the strength of the evidence is insufficient or inappropriate. (Line 281-283)

2. There is a risk of over-interpretation by presuming the interaction between ligand compounds (three short-chain fatty acids) and NADPH oxidase and iNOS with the values in Table 3. This manuscript does not have proper references to corroborate, especially the significantly low EFE (free energy value). In addition, the analysis results did not present RMSD values, hydrogen bonds between protein-ligands and hydrophobic interactions (Ligplot+ analysis).

3. The manuscript emphasizes that fecal SCFA is essential in treating CRC. Therefore, the raw data drawn in Figure 6 should be attached in the Supplementary Information.

Minor suggestions:

1. The references cited in the manuscript are old and should be updated.

2. Incomprehensible. Line 262, "Figures 2B and 6".

3. Lines 331-332. As shown in Figure 3, it should be IL-10, not IL-12. ”….both showed a positive correlation with fecal pH, TNF, and IL-12 (p<0.05).”

Author Response

Manuscript ID: nutrients-2474912

# Reviewer 1

This manuscript demonstrates that YF, a source of fructooligosaccharides, can reduce inflammatory biomarkers in CRC-induced animals. And the test results confirmed that YF could increase the secretion of sIgA and speculated the interaction between SCFA and NADPH oxidase and iNOS through in silico analysis. Although the animal experiments in the manuscript are adequately designed, and the text is clear and easy to read, some data are not measured or presented, so the credibility of the experimental results of the manuscript is slightly weak. For example, how much SCFA is produced by YF treatment? It is too exaggerated to use SCFA as an enzyme docking analysis to speculate on reducing cancer biomarkers. Some revisions will be more suitable for publication.

Author’s response: Thanks for your comments, which greatly added to our work. Some of the data used in this work were previously published (Grancieri et al., 2017), which is why we did not repeat the raw results in this manuscript, but only used them for statistical correlation. In our previous work, we found that the Y and CY groups had greater production of SCFA acetate, propionate, and butyrate, a fact that served as the basis for carrying out the in silico and correlation analyses of the present manuscript. However, as requested, we have added the raw data to the supplemental material (Table S1).

Reference used:

Grancieri, M., Costa, N. M. B., Tostes, M. D. G. V., de Oliveira, D. S., de Carvalho Nunes, L., de Nadai Marcon, L., ... & Viana, M. L. (2017). Yacon flour (Smallanthus sonchifolius) attenuates intestinal morbidity in rats with colon cancer. Journal of Functional Foods37, 666-675.

Main recommendations:

  1. In Figure 3, the endotoxin of the CY group is compared with that of the C, Y, and S groups, which can be regarded as ambiguous. Extrapolating from this data that consumption of yacon may reduce serum inflammation by reducing intestinal permeability, bacterial translocation (by reducing lipopolysaccharide release), and inducing inflammatory cytokines, the strength of the evidence is insufficient or inappropriate (Line 281-283).

Author’s response: Thank you for your comment.  We made this statement based on the fact that animals, even with colorectal cancer (CY group), have endotoxemia levels similar to healthy animals and we associated the results with an effect of yacon. However, we agree that the statement extrapolates the results, so changes were made to the title (line 3), discussion (line 382), and conclusion (line 456) of the manuscript.

  1. There is a risk of over-interpretation by presuming the interaction between ligand compounds (three short-chain fatty acids) and NADPH oxidase and iNOS with the values in Table 3. This manuscript does not have proper references to corroborate, especially the significantly low EFE (free energy value). In addition, the analysis results did not present RMSD values, hydrogen bonds between protein-ligands, and hydrophobic interactions (Ligplot+ analysis).

Author’s response: Thank you for your comment. We agree with the comment and have added a description of the types of bonds between ligand-molecule in the figure caption (please check lines 322-323). Furthermore, based on the assertions we make of protein-binding interaction are based on the literature which states that when the change of the system free energy is negative can the protein–ligand binding occurs spontaneously and it is even stated that bonds greater than -4 kcal.mol-1 are considered very strong (Du et al., 2016) and these were precisely observed for iNOS and NADPH oxidase with butyrate.

Reference used:

Du, X., Li, Y., Xia, Y. L., Ai, S. M., Liang, J., Sang, P., ... & Liu, S. Q. (2016). Insights into protein–ligand interactions: mechanisms, models, and methods. International journal of molecular sciences17(2), 144.

  1. The manuscript emphasizes that fecal SCFA is essential in treating CRC. Therefore, the raw data drawn in Figure 6 should be attached in the Supplementary Information.

 Author’s response: Thank you very much for your comment. We agree with the reviewer and have added the raw data to the supplementary material (Table S1).

Minor suggestions:

  1. The references cited in the manuscript are old and should be updated.

Author’s response: Thank you for your comment. The references were updated.

  1. Incomprehensible. Line 262, "Figures 2B and 6".

Author’s response: Thank you very much for your comment. This typing error has been fixed.

  1. Lines 331-332. As shown in Figure 3, it should be IL-10, not IL-12. ”….both showed a positive correlation with fecal pH, TNF, and IL-12 (p<0.05).”

Author’s response: Thank you very much for your comment. The results have been correctly rewritten (please, see lines 328-329).

Reviewer 2 Report

The study of Mariana Grancieri et al. aimed to investigate the effects of Yacon flour against oxidative stress, inflammation, and endotoxemia in rats with induced colorectal cancer through in vivo and in silico assays.

The manuscript is well written, informative to readers, and in accordance with the aim and scope of Nutrients.

However, there is some information that needs to improve to make the manuscript more informative, including:

1. Section 2.10: the reference of target enzymes used in this study is required.

Also, the information about the active site of target enzymes, the binding mechanism of docked ligands, and key binding interactions should be discussed in the Discussion section of this study.

Figure 5 should be revised to have a larger size and enhanced resolution, making it easier for the reader to view.

2. There are some small typos need to revise:

-Line 27: reduce CRC-damages.the damages caused by colorectal cancer

Author Response

Manuscript ID: nutrients-2474912

 # Reviewer 2

The study of Mariana Grancieri et al. aimed to investigate the effects of Yacon flour against oxidative stress, inflammation, and endotoxemia in rats with induced colorectal cancer through in vivo and in silico assays. The manuscript is well written, informative to readers, and in accordance with the aim and scope of Nutrients. However, there is some information that needs to improve to make the manuscript more informative, including:

Author’s response: Thank you very much for your comments which helped a lot to improve our article.

  1. Section 2.10: the reference of target enzymes used in this study is required. Also, the information about the active site of target enzymes, the binding mechanism of docked ligands, and key binding interactions should be discussed in the Discussion section of this study. Figure 5 should be revised to have a larger size and enhanced resolution, making it easier for the reader to view.

Author’s response: Thank you for your suggestion. The active site of target enzymes was based on an exhaustive search in the literature of research that demonstrated the binding sites of the enzymes, then based on these results, successive tests were carried out in order to locate the best active site of the enzymes. This information was added on lines 233-236. Furthermore, the size of the figure was increased to better identify the types of links and link residues (Figure 5). In the figure caption, the meaning of the colors of the binding residues was added and these were further discussed (please see lines 406-409).

  1. There are some small typos need to revise:

-Line 27: reduce CRC-damages. The damages caused by colorectal cancer

Author’s response: Thank you for your comment. The sentence was rewritten. Please, see lines 26-27.

Round 2

Reviewer 1 Report

Molecular docking simulation is only one of the reasonable speculations using in silico analysis to speculate on possible phenomena, and it is not ironclad proof. To confirm the fact of docking between SCFA and the candidate enzymes, in fact, it is necessary to make a point mutation in the nucleic acid sequence expressing the protein (enzyme), that is, to change the amino acid at the position of docking SCFA. Since the text uses a direct and assertive way to express the regulatory mechanism of SCFA on the candidate enzymes, and then locates the physiological role of YF on animal pathological models.

 In addition, the lower the mean binding (kcal/mol), the tighter the binding between ligands (such as SCFAs) and proteins (candidate enzymes).

 Based on the above reasons, the reviewers still strongly recommend that the authors of this manuscript should be more cautious in processing the information in this part of the in silico analysis.

Author Response

# Reviewer

Molecular docking simulation is only one of the reasonable speculations using in silico analysis to speculate on possible phenomena, and it is not ironclad proof. To confirm the fact of docking between SCFA and the candidate enzymes, in fact, it is necessary to make a point mutation in the nucleic acid sequence expressing the protein (enzyme), that is, to change the amino acid at the position of docking SCFA. Since the text uses a direct and assertive way to express the regulatory mechanism of SCFA on the candidate enzymes, and then locates the physiological role of YF on animal pathological models.

In addition, the lower the mean binding (kcal/mol), the tighter the binding between ligands (such as SCFAs) and proteins (candidate enzymes).

Based on the above reasons, the reviewers still strongly recommend that the authors of this manuscript should be more cautious in processing the information in this part of the in silico analysis.

Author’s response: We appreciate your consideration. We have modified the sentences to be more careful in asserting these bindings in silico (please, see lines 395-396 and 409). In addition, we have added a sentence at the end of the discussion highlighting this caution in asserting the bindings between SCFAs and enzymes “However, we emphasize that these connections between SCFA and enzymes, demonstrated in the in silico analysis, are plausible to occur, but require caution in their interpretation, since in a physiological context, other factors can interfere in these connections and in the consequent actions of the enzymes [54]”, please, see pages 443-446.